# Comparison of Subjective Facial Emotion Recognition and “Facial Emotion Recognition Based on Multi-Task Cascaded Convolutional Network Face Detection” between Patients with Schizophrenia and Healthy Participants

**DOI:** 10.3390/healthcare10122363

**Published:** 2022-11-24

**Authors:** Toshiya Akiyama, Kazuyuki Matsumoto, Kyoko Osaka, Ryuichi Tanioka, Feni Betriana, Yueren Zhao, Yoshihiro Kai, Misao Miyagawa, Yuko Yasuhara, Hirokazu Ito, Gil Soriano, Tetsuya Tanioka

**Affiliations:** 1Graduate School of Health Sciences, Tokushima University, Tokushima 770-8509, Japan; 2Graduate School of Engineering, Tokushima University, Tokushima 770-8506, Japan; 3Department of Psychiatric Nursing, Nursing Course of Kochi Medical School, Kochi University, Kochi 783-8505, Japan; 4Department of Physical Therapy, Hiroshima Cosmopolitan University, Hiroshima 734-0014, Japan; 5Rozzano Locsin Institute, Tokushima 770-8509, Japan; 6Department of Psychiatry, Fujita Health University, Nagoya 470-1192, Japan; 7Department of Mechanical Engineering, Tokai University, Tokyo 151-8677, Japan; 8Department of Nursing, Faculty of Health and Welfare, Tokushima Bunri University, Tokushima 770-8514, Japan; 9Institute of Biomedical Sciences, Tokushima University, Tokushima 770-8509, Japan; 10Department of Nursing, College of Allied Health, National University Philippines, Manila 1008, Philippines

**Keywords:** facial emotion recognition, human–robot interaction, multi-task cascaded convolutional networks, reliability, patients with schizophrenia, healthy participants

## Abstract

Patients with schizophrenia may exhibit a flat affect and poor facial expressions. This study aimed to compare subjective facial emotion recognition (FER) and FER based on multi-task cascaded convolutional network (MTCNN) face detection in 31 patients with schizophrenia (patient group) and 40 healthy participants (healthy participant group). A Pepper Robot was used to converse with the 71 aforementioned participants; these conversations were recorded on video. Subjective FER (assigned by medical experts based on video recordings) and FER based on MTCNN face detection was used to understand facial expressions during conversations. This study confirmed the discriminant accuracy of the FER based on MTCNN face detection. The analysis of the smiles of healthy participants revealed that the kappa coefficients of subjective FER (by six examiners) and FER based on MTCNN face detection concurred (κ = 0.63). The perfect agreement rate between the subjective FER (by three medical experts) and FER based on MTCNN face detection in the patient, and healthy participant groups were analyzed using Fisher’s exact probability test where no significant difference was observed (*p* = 0.72). The validity and reliability were assessed by comparing the subjective FER and FER based on MTCNN face detection. The reliability coefficient of FER based on MTCNN face detection was low for both the patient and healthy participant groups.

## 1. Introduction

The symptoms of patients with schizophrenia can be divided into three categories: positive symptoms (the emergence of conditions that do not exist when the patient is healthy), negative symptoms (the loss of conditions that exist when the patient is healthy), and the impairment of social cognitive functions (including memory and attention disorders) [1]. The most common positive symptoms include hallucinations and delusions; among these, auditory hallucinations are the most common. Moreover, due to the impaired processing of negative verbal information, symptoms, such as sudden excitement and screaming, are observed [2]. Typical negative symptoms include avoidance, apathy, emotional numbness [3], and unsociability [4], including indifference to one’s surroundings. Additionally, social cognitive dysfunctions manifest as an impaired ability to selectively pay attention (select and pay attention to information), compare and contrast (compare with memories), and form concepts (classify and conceptualize things) [4], which may lead to difficulties in daily life [4]. Many patients are unaware of their illness, and these symptoms may interfere with their mood, behavior, and relationships [5].

The number of patients with psychiatric disorders, including schizophrenia, is increasing in Japan. Furthermore, the shortage of nurses and care workers in chronic-illness care wards is a problem owing to the aging population and extremely low birthrate. Therefore, social support robots have been recently developed to observe the health status of older adults and manage their health [6]. The introduction of various robotic technologies, including communication robots, therapy robots, and monitoring sensors, has been initiated in developed countries that have advanced medical environments [6,7,8].

Introducing patients with schizophrenia to robotic technology has proven difficult due to the following reasons: (1) these patients have communication-related issues and (2) unlike healthy people, they have a poor perception of facial emotional expressions [9,10]. This difficulty could also partly be attributed to the fact that robot-assisted psychiatric rehabilitation is not covered by health insurance. A study examined the sensory perception of patients with schizophrenia using the humanoid robot Nao and found that these patients could maintain interpersonal cooperation during interactions with the robot [11].

Communication problems are considered a primary clinical pathology in patients with schizophrenia [12], who sometimes struggle to understand and communicate their own feelings as well as those of others. The social lives of these patients are significantly affected by their inability to process social information and adjust their interpersonal behaviors accordingly [5,13]. Furthermore, the unsociability of patients with schizophrenia is another problem, and its negative effects on interpersonal interactions have been investigated [14]. Nonetheless, compared to studies on patients with autism or dementia, fewer studies have focused on patients with schizophrenia. Furthermore, there are limited examples in the literature that is focused on analyzing the facial expressions of patients with schizophrenia with the aim of helping others understand their emotions.

Thus, the accurate detection of the emotions of patients with schizophrenia during interactions using facial emotion recognition (FER) by robots enabled by the multi-task cascaded convolutional network (MTCNN) face detection function could facilitate the early detection of changes in the patient’s symptoms.

Therefore, it is essential to investigate how patients with schizophrenia communicate with robots to develop a support service robot for them [15]. The ability of a healthcare robot to determine a patient’s facial expression and accurately convey it to others is essential. Such FER for robots might be able to decrease the risk of communication-related errors among the patients, healthcare robots, and healthcare providers, as well as the likelihood that the patient will experience stress as a result of their inability to express emotions. This study aimed to compare the results of subjective FER and FER based on the MTCNN face detection between patients with schizophrenia and healthy participants.

## 2. Materials and Methods

### 2.1. Research Design

This study was designed in accordance with the intentional observational clinical research design (IOCRD) [16], which simultaneously generates qualitative and quantitative data through specialized observation and measurement processes using advanced technical equipment. Figure 1 shows the measurement and analysis tools used in this study. Qualitative data consisted of field notes, digital video recording, and subjective FER by three examiners, whereas quantitative data consisted of FER based on MTCNN face detection.

The measurement timelines in the observational (qualitative) and experimental (quantitative) studies were synchronized using radio clocks to achieve the simultaneous research data collection. The alignment of the timeframes is crucial because the frame-by-frame analysis by MTCNN might lead to inconsistent findings otherwise. A stationary picture (frame) that constitutes a moving image is referred to as a frame. The number of frames that are rewritten per second is expressed in “frames per second” (fps), which is used to measure how smoothly the video plays.

The MTCNN framework proposed by Zhang et al. [17] was used in the study, wherein face detection and alignment were integrated using unified CNNs with multi-task learning.

### 2.2. Participants

The study initially included 82 participants (age: 40–80 years) who met the inclusion criteria; these comprised 41 patients with schizophrenia and 41 age-matched healthy volunteers.

The participants were selected on the basis of two criteria: (1) patients without any change in the medication dosage for schizophrenia or other disorders and (2) patients who had received stable treatment and experienced psychiatric symptoms for the past 3 months. Both patients and healthy participants could converse in Japanese. The exclusion criteria included treatment for alcohol or drug (stimulant) dependence, aggressive or violent behavior, residual stage of schizophrenia, the presence of severe comorbidities, severe hearing impairment, an inability to understand Japanese, and inability to provide informed consent.

After excluding the participants with missing values in the interim analysis, 71 participants were enrolled in this study; these comprised 31 patients with schizophrenia (patient group) and 40 healthy individuals (healthy participant group). This number includes the 10 participants in the preliminary experiment.

### 2.3. Data Collection

The day and conference rooms in the wards of a psychiatric hospital were used as the study’s experimental venues. The Pepper Robot (SoftBank Robotics Corp., Tokyo, Japan) was used in this study; it was equipped with a dialog application program that was jointly developed by the Tanioka research team and the Xing Company [18]. This program facilitated empathetic conversations between the participants and Pepper and enabled the use of a tablet PC and keyboard to remotely control Pepper’s speech and head motion. In a separate room, an operator (aided by an assistant) remotely input Pepper’s actions, such as acknowledgment gestures and nodding, during the conversation [18]. The conversation between the participants and Pepper lasted for approximately 10 min and was recorded using two digital video cameras. One examiner contacted the operator to adjust the end time and recorded the important events that occurred during the conversation as well as the time on the radio clock in a field note [19,20].

Figure 2 shows the environmental setup, wherein a wall or a barrier is placed behind the participants to prevent errors in FER based on MTCNN face detection. The distance between the participants and Pepper was set to 84 cm, considering the detection range of Pepper’s infrared sensor and the range of motion of its upper limbs. The first video camera was set in a position where the participants could see the robot’s facial expressions with their eyes, while the second video camera was placed in a position where both the robot and the participant’s facial expressions were visible. The distances between Pepper and the first and second video cameras were 1.5 m and 2.15 m, respectively. The participants were asked to remove their masks to enable the easier reading of their facial expressions. Data were collected between December 2020 and November 2021.

### 2.4. Analysis Methods

FER comprised two types, subjective FER (by three medical experts) and FER based on MTCNN face detection.

#### 2.4.1. Subjective FER by Medical Experts

Because FER can significantly differ depending on the age and sex of the examiners [21,22], three examiners were selected to ensure fairness. These comprised Nurse A (a man with less than 10 years of experience), Nurse B (a woman with more than 10 years of experience), and a psychiatrist (a man with more than 10 years of experience). The FER based on the MTCNN face detection is an analytical tool that assesses facial expressions from videos and images; in accordance with the standard protocol, Nurse A reviewed the video data without audio. To evaluate the examinees’ facial expressions as objectively as possible, a drop-down list similar to that of FER based on MTCNN face detection was created; it comprised the following seven categories: angry, disgusted, fearful, pleased, sad, surprised, and neutral. There was a possibility of data misalignment, and the outcomes would have differed if each of the three examiners had watched the entire video and only cut out instances when there was a change in facial expressions. Therefore, we cut out 5–10 s of the video using Shotcut (Dun & Bradstreet, Inc., Stockholm, Sweden) while focusing on the area of interest at the point where Nurse A determined that there was a change in the facial expression; this served as the criterion for judgment. These cut videos showed a portion of the conversation from just before the participants shifted their facial expressions to the change in facial expression. One data set per participant was used. The psychiatrist and Nurse B assessed 71 test participants whose data were usable without any missing values. The facial expressions in the same area of interest were solely examined using audio-less facial expression data under similar circumstances as that of Nurse A to strengthen the validity of the examiner’s subjective FER.

#### 2.4.2. Analysis by the FER Based on MTCNN Face Detection Algorithm

Research in the field of affective computing has focused on detecting and classifying human emotions from facial expressions, and advances in deep learning have increased the number and accuracy of methods for detecting facial emotions [23]. The multi-task EfficientNet-B2 achieved 66.29% accuracy with 7 emotions [24].

However, we are considering the use of a relatively small network to run on small PCs and robots in the future; therefore, our research group decided to use a combination of MTCNN-based face detection and DCNN (deep convolutional neural networks)-based expression recognition models.

Zhang et al. used MTCNN to jointly perform face detection and alignment by coarsely to finely predicting the positions of faces and landmarks [17]. Consequently, the accuracy of real-time face detection and positioning improved, MTCNN had the best comprehensive performance in face recognition [25], and MTCNN enabled rapid face detection. The trained method [26] can classify facial expressions after face detection with the MTCNN [27]. Therefore, in our study, MTCNN was used for highly accurate face detection.

Subsequently, FER was performed using an expression of the classification model; the expressions analyzed were anger, disgust, fear, joy, sadness, surprise, and neutral. The video analysis output was set to 30 fps, thereby producing 30 results. The facial expression judgment result was presented in relation to the aforementioned seven facial expressions so that the total score was 100; the result with the highest mean value was used.

#### 2.4.3. Establishing a Region of Interest

In this study, Nurse A reviewed all the videos that lasted for approximately 10 min, and the regions of interest were defined as those that lasted between 5 and 10 s and comprised emotional changes during the participant’s conversation with Pepper as well as the FER based on MTCNN face detection findings. The region of interest describes how JPEG 2000 compresses pictures by allocating a significant amount of code to a specific region. The term “region of interest” refers to a particular area that has been narrowed down for observation or measurement by using several imaging techniques. Shotcut was used as the video-editing software, and the video size was adjusted to 640 widths, 480 frames, and 29.97 fps.

#### 2.4.4. Preliminary Experiment to Confirm Discriminant Accuracy of the FER Based on MTCNN Face Detection

A preliminary experiment was conducted in order to confirm the discrimination accuracy of the FER based on MTCNN face detection. In this experiment, six healthy examiners (age: 30–60 years) were asked to evaluate a video in which a subject obviously looked happy; their findings were compared with the evaluation result “happy” obtained from the FER based on MTCNN face detection. Examiner A was excluded from these examiners.

The videos were shown to each examiner separately to prevent any influence on their opinions. Furthermore, to match the assessment conditions with those of the FER based on MTCNN face detection, the participants were asked to evaluate the videos without audio and to select the applicable items from the seven items used in the FER based on the MTCNN face detection.

#### 2.4.5. Statistical Processing Method

In the preliminary experiment, the agreement rate and kappa (*κ*) coefficient between the subjective FER (evaluated by the aforementioned six healthy examiners) and FER based on MTCNN face detection were calculated.

In the actual experiment, Welch’s *t*-test was used to confirm the absence of a difference in the mean age between the patient and healthy participant groups. A perfect agreement was defined as an agreement between the subjective FER (evaluated by the aforementioned six healthy examiners) and FER based on MTCNN face detection results from examiners A, B, and C. Fisher’s exact probability test was used to determine the agreement rate.

To confirm the variance of the facial expression judgment results from the three examiners and from the FER based on MTCNN face detection, an *F*-test was used with a true value of 0. In a two-way variational effect model, when both the human effect and the measurement effect were included as variables, we compared the means of the three examiners’ subjective FER and the FER based on the MTCNN face detection’s intraclass correlation coefficient (ICC) and obtained 95% confidence interval (CI) measurements in the patient and healthy participant groups. Cronbach’s alpha coefficient was used to confirm the internal consistency. In this study, a *p*-value of less than 0.05 was considered statistically significant.

IBM SPSS Statistics for Windows, Version 27.0. (Armonk, NY: IBM Corp) was used for the statistical processing of the data.

### 2.5. Ethical Considerations

This study was approved by the Ethics Committee of the Tokushima University Hospital (#3046) and the Mifune Hospital Clinical Research Ethics Review Committee (#20180502).

## 3. Results

The mean ages of the patient and healthy participant groups were 56.13 ± 11.45 years and 57.69 ± 9.5 years, respectively; no significant intergroup differences were observed in the age (*t* = 0.61, *p* = 0.54).

### 3.1. Reliability and Validity of the FER Based on MTCNN Face Detection

Table 1 shows the level of agreement and the kappa coefficients between the subjective FER from the six aforementioned examiners for the videos of 10 healthy participants (included in the actual experiment) who were subjectively evaluated as “smiling” by Examiner A and FER based on MTCNN face detection. The kappa coefficient was 0.63 (*p* = 0.003).

### 3.2. Percentage of Agreement between the Subjective FER and FER Based on MTCNN Face Detection

Table 2 shows the percentage of agreement between the subjective FER (determined by examiners A, B, and C) and the FER based on MTCNN face detection in the patient and healthy participant groups.

The perfect agreement rate between the subjective FER and FER based on MTCNN face detection was 3/31 (9.68%) for the patient group and 6/40 (15.00%) for the healthy participant group.

The agreement rates between the subjective FER and FER based on MTCNN face detection were as follows: (a) patient group: Examiner A = 40.63%, Examiner B = 34.38%, and Examiner C = 25.00%; (b) healthy participant group: Examiner A = 42.50%, Examiner B = 52.50% and Examiner C = 17.50%.

Table 3 shows the results of Fisher’s exact probability test for the agreement between the subjective FER and FER based on MTCNN face detection in the patient and healthy participant groups (exact *p*-value [two-tailed]: 0.72).

### 3.3. Reliability of the FER

Table 4 shows a comparison of the means of the subjective FER from the three examiners and the FER based on the MTCNN face detection’s ICC and 95% CI measurements in the patient and healthy participant groups. The ICC (2, 1) in the patient group was 0.41 (95% CI, −0.02 to 0.69; F = 1.7; *p* = 0.03), with a Cronbach’s alpha coefficient of 0.41. Alternatively, the ICC in the healthy participant group was 0.24 (95% CI, −0.24 to 0.56; F = 1.31; *p* = 0.13), with a Cronbach’s α coefficient of 0.24.

## 4. Discussion

For facial expressions that were considered happy by everyone, the kappa (*κ*) coefficient was 0.63 (Table 1). A high *κ* value indicated a good level of agreement among the examiners. Thus, the results were considered practically consistent, and the FER based on MTCNN face detection was considered generally capable of recognizing salient facial expressions. However, during the subjective FER for healthy participant #3 (Table 1), two of the six examiners (examiners #5 and #6) evaluated the facial expression as “surprised” because the corners of the participant’s mouth were raised at all times. Moreover, the subjective result for healthy participant #6 in Table 1 differed from the FER based on the MTCNN face detection result (which was “fear”). Differences from the other videos were considered because the participants talked while tilting their heads at an angle, and their glasses sometimes reflected blue due to blue light.

The agreement rate between each examiner’s subjective FER and FER based on MTCNN face detection was low; furthermore, the perfect agreement rate among examiners A, B, and C was very low. This study evaluated the examiners’ subjective FER as objectively as possible; however, the agreement rate decreased because of the possibility that each examiner’s FER might differ (Table 2). For example, a smile may include positive facial expressions (such as a broad smile or a smiling face) or negative facial expressions (such as an affectionate smile or a bitter smile). Thus, it was considered that both the FER results differed.

The comparison between the three examiners’ subjective FER and the FER based on the MTCNN face detection results in the patient and healthy participant groups revealed no significant differences in the level of agreement (Table 3). Women were reportedly more accurate and faster at recognizing facial expressions than men [21]. Two of the three examiners were men, and one was a woman, which may have resulted in the differences in the reading of the facial expressions. Furthermore, the facial expression recognition test (which was developed using a two-parameter logistic model of the item-response theory) revealed gender-based differences in the ability to recognize facial expressions and an age-related decrease in the facial expression recognition ability, suggesting that differences in facial expression recognition are dependent on the age of the examiners. One study noted that the ability to recognize facial expressions differed depending on the age of the examiners [28]. Therefore, it was considered difficult to evaluate similar facial expressions, such as sadness and dissatisfaction, as the evaluation of these expressions tended to vary.

The mean ICC (2, 1) for the FER based on MTCNN face detection in the patient group was somewhat consistent, but the reliability was considered low. The mean ICC (2, 1) for the subjective FER (reported by the three examiners) and FER based on the MTCNN face detection results in both the patient and healthy participant groups were low; the reliability coefficients were also low. These findings suggest that the results are unreliable (Table 4).

In a study wherein facial expressions were evaluated in patients with schizophrenia and healthy participants, spontaneous facial expressions in patients with schizophrenia did not necessarily reflect their internal emotions. For example, the greater the negative symptoms (e.g., emotional numbing and indifference) demonstrated by the patients with schizophrenia, the more difficult it was to classify their facial expressions [29]. In addition, in patients with schizophrenia with severe negative symptoms, angry and fearful expressions tend to be misclassified as “neutral”, and there is a negative correlation between negative symptoms and accurate classification of angry expressions. Furthermore, patients with schizophrenia who score higher on emotional aspects (depression, anxiety, etc.) tended to conflate joy and anger [30]. Thus, several factors may affect facial expressions as well as their correct identification in patients with schizophrenia.

Several studies have established that internal facial features (i.e., eyes, nose, and mouth) and their surrounding regions transmit diagnostic cues for expression recognition [31,32] and that people frequently focus more on local facial regions that are the most distinctive for each expression (such as the mouth in happy faces and the eyes in sad and fearful faces) [33,34,35]. With this, it is possible that in order to accurately interpret low-intensity facial emotions, we rely on cues from other facial areas to confirm the cues from the “diagnostic” regions when the expressive facial signals are faint or unclear [36].

According to Darwin’s universal theory, six basic internal human emotions are expressed using the same facial movements in all cultures; this is consistent with the results of FER based on MTCNN face detection in a previous study [37]. Unlike oriental people, Westerners express each of the six basic facial expressions using different sets of facial muscles [37]; this is especially true for surprise, fear, disgust, and anger, which are characterized by a considerable overlap among facial expressions [38]. The preliminary experiment revealed an agreement between the subjective FER and the FER based on the MTCNN face detection in the case of smiles. This agreement, however, was not observed in the actual experiment in both the healthy participant and patient groups. Thus, we considered that the six basic facial expressions were not well expressed among the Japanese participants.

The reduced reliability of the FER based on the MTCNN face detection findings for Asians compared with that for Westerners was attributable to the ethnic proportions in the deep-learning database [39]. In our study, as shown in Table 2 and Table 3, there were discrepancies in the subjective FER and FER based on the MTCNN face detection for each frame between the patient and healthy participant groups. Our findings suggested that both humans and FER based on the MTCNN face detection evaluation of facial expressions is difficult, except in the case of facial expressions that are clearly recognized by humans as happy.

### Limitations of the Study

The discrepancy between the data used to train face detection and the facial expression evaluation models in the FER based on MTCNN face detection and the image data used in the experiment may be a cause of the FER based on MTCNN face detection’s inability to produce analytical results. This disparity may be caused by the heavy reliance on facial expression databases on artificial (rather than natural) facial expressions. Furthermore, our findings have limited generalizability.

## 5. Conclusions

This preliminary study confirmed the accuracy of FER based on MTCNN face detection in differentiating facial expressions that were considered “happy” by all examiners. These results were substantially consistent. The validity and reliability were assessed by comparing the subjective FER and FER based on MTCNN face detection. The reliability coefficient of FER based on MTCNN face detection was low for both the patient and healthy participant groups. Furthermore, the reliability of the FER based on MTCNN face detection during conversations with the robot was low for both patients and healthy participants. The ability of a healthcare robot to determine a patient’s facial expression and convey it to others is essential. Thus, further research is needed to develop the robot’s function to determine the patients’ emotions from their facial expressions.

## Figures and Tables

**Figure 1 healthcare-10-02363-f001:**
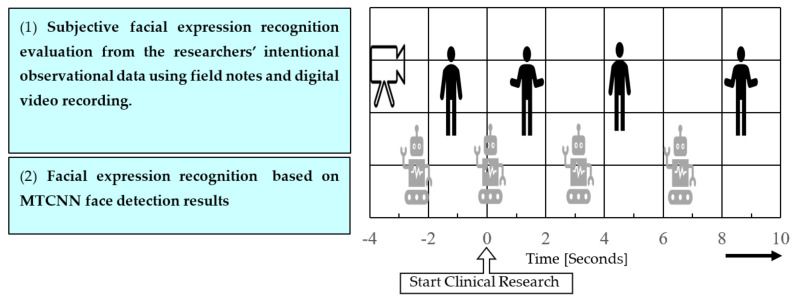
Research framework simultaneous research data collection designed in accordance with the intentional observational clinical research design [16]. Arrow indicates timeline.

**Figure 2 healthcare-10-02363-f002:**
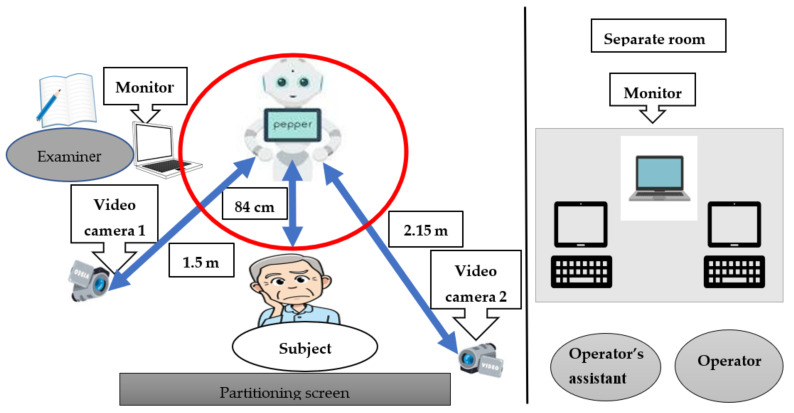
Experimental environment. The blue arrows indicate the distance from the Pepper robot to the digital video camera.

**Table 1 healthcare-10-02363-t001:** Agreement rate and kappa coefficients between the subjective FER from the six examiners for the smiles of 10 healthy participants and FER based on MTCNN face detection.

Healthy Participants	Age (Years)	FER Based on MTCNN FaceDetection	Examiner 1	Examiner 2	Examiner 3	Examiner 4	Examiner 5	Examiner 6	Coefficient of Identity (%)
1	40	Happy	Neutral	Happy	Neutral	Happy	Happy	Happy	71.43
2	50	Happy	Neutral	Happy	Neutral	Happy	Happy	Happy	71.43
3	60	Happy	Happy	Happy	Happy	Happy	Surprised	Surprised	71.43
4	60	Happy	Happy	Happy	Happy	Happy	Happy	Neutral	85.71
5	60	Happy	Neutral	Neutral	Neutral	Happy	Happy	Happy	42.86
6	60	Fear	Neutral	Happy	Neutral	Happy	Happy	Happy	42.86
7	60	Happy	Neutral	Happy	Neutral	Happy	Happy	Happy	71.43
8	70	Happy	Neutral	Happy	Happy	Happy	Happy	Happy	85.71
9	70	Happy	Happy	Happy	Happy	Happy	Happy	Neutral	85.71
10	70	Happy	Happy	Happy	Surprised	Happy	Happy	Surprised	71.43

The results of the FER are shown. The kappa coefficient (κ) is 0.63. The coefficient of identity (%) indicates the percentage agreement between the subjective FER for each examiner’s ratings and FER based on MTCNN face detection. FER—facial emotion recognition, MTCNN—multi-task cascaded convolutional network.

**Table 2 healthcare-10-02363-t002:** Comparison of the agreement between the subjective FER reported by the three examiners and the FER based on MTCNN face detection in the patient and healthy participant groups.

Participants	Examiner A	Examiner B	Examiner C
(a) Patient group	40.63 (%)	34.38 (%)	25.00 (%)
(b) Healthy participant group	42.50 (%)	52.50 (%)	17.50 (%)

The perfect coincidence ratio between the subjective FER for all three examiners and the FER based on MTCNN face detection was 3/31 (9.68%) in the patient group and 6/40 (15.00%) in the healthy participant group. FER—facial emotion recognition, MTCNN—multi-task cascaded convolutional Network.

**Table 3 healthcare-10-02363-t003:** Agreement between subjective FER and FER based on MTCNN face detection in the patient and healthy participant groups.

Participants	Not Matched	Matched
(a) Patient group (n = 31)	28	3
Adjusted residual	0.7	−0.7
(b) Healthy participant group (n = 40)	34	6
Adjusted residual	−0.7	0.7

Exact *p*-value (two-tailed) calculated by the Fisher’s exact probability test = 0.72.

**Table 4 healthcare-10-02363-t004:** Comparison of the means of the subjective FER from the three examiners and the FER based on MTCNN face detection’s ICC and 95% CI measurements in the patient and healthy participant groups.

Participants			ICC	95% CI		F-Test with True Value of 0			
				Lower Limit	Upper Limit	Value	Degree of Freedom 1	Degree of Freedom 2	*p*-Value
(a) Patient group	FER based on MTCNN face detection	Average Measured Value	0.41	−0.02	0.69	1.70	30.00	90.00	0.03
(b) Healthy participant group			0.24	−0.24	0.56	1.31	39.00	117.00	0.13

Binary-variate effect models when both the human and measured effects are variates. (a) Cronbach’s α = 0.41 for FER based on MTCNN face detection (patient group). (b) Cronbach’s α = 0.24 for FER based on MTCNN face detection (healthy participant group).

## Data Availability

Data presented in this study are available upon request from the corresponding author. Data are not publicly available because of privacy and ethical restrictions.

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
