# Peer review of "Comparison of Subjective Facial Emotion Recognition and “Facial Emotion Recognition Based on Multi-Task Cascaded Convolutional Network Face Detection” between Patients with Schizophrenia and Healthy Participants"

_healthcare, 2022, doi:10.3390/healthcare10122363_

Round 1
Reviewer 1 Report
In the manuscript, the authors conducted an experiment to investigate the reliability and applicability of multi-task cascaded convolutional networks (MTCNN) to facial emotion recognition of schizophrenia patients. In the preliminary experiment, the authors confirmed its capability by comparing its prediction with those by 6 human examiners using videos of smiling healthy participants. However, predictions by MTCNN and 3 human examiners were not in good agreement both in the patient and control groups in the experiment using videos of 31 schizophrenia patients and 40 healthy controls during conversation. Though the experiments were well performed, there are several major issues to be addressed in the current manuscript.
The authors did not provide sufficient information on the MTCNN system utilized in the study. The source of the MTCNN need to be mentioned in the manuscript for reproducibility. If it was downloaded from specific webpages, the URLs should be added. Moreover, the original paper (Zhang et al., 2016) proposed MTCNN as a face detection and alignment system, and facial emotion recognition is not included in it. The authors should explain the architecture of the facial emotion recognition system as well as the reason for the selection of 7 emotion categories used in the analysis.
The authors used MTCNN as the facial emotion recognition method. However, no statements are included in the manuscript about the reasons for the selection of MTCNN. The authors should describe the reasons and its advantages for the usage of MTCNN. Moreover, it should be mentioned whether methods other than MTCNN can outperform it and attain ratings comparable to human examiners in the task in the discussion.
In the preliminary experiment, the authors treat each human examiner as a rater and used their predictions in the statistical tests without processing. However, it sounds unnatural that statistical tests include predictions by multiple human examiners and only one MTCNN when comparing MTCNN-based and human-based predictions, because agreement/disagreement of predictions among human examiners would affect the result. Instead, the authors should calculate representative human predictions by max voting or other methods and compare them with those by MTCNN. Comparison between the subjective FER and FER of MTCNN using perfect agreement rate in the experiment should be revised as well.
The results of the preliminary experiment seem to contradict those of the experiment. The authors examined the validity of MTCNN in 10 healthy participants and showed good agreement among MTCNN and human examiners in the preliminary experiment. However, in the experiment the predictions of MTCNN and human examiners were not in good agreement even in the recognition of the healthy participants’ facial expression. The authors should suggest any reasons for the inconsistency.
The authors compared the validity of MTCNN between facial emotion recognition of patients and of healthy controls. However, the reasons for the absence of difference were not clearly discussed. The authors should add discussion on them in the manuscript.
Minor issues:
The original paper that proposed MTCNN should be cited.
The authors should clarify whether the ten healthy participants who joined the preliminary experiment were also included in the experiment.
It is not clear how many segments of 5 to 10 seconds were used in the analysis per participant.
The authors used part of video where facial expression changed. Which part of the videos did the examiners use to determine the category of the facial expression, before or after the change of the facial expression?
In section 2.4.5. Statistical Processing Method, statistical procedures for the experiment and those for the preliminary one should be described separately for clarity.
In the preliminary experiment, p-value of the kappa coefficient should be added.
In line 264, the meaning of “mean FMR” is not clear, because FMR is used without explanation.
In the paragraph beginning in line 322, the authors attribute the difficulty in expressing natural expressions by the participants to sleep deprivation and stressful environments. However, the papers [26] and [27] are not about their effects on the facial expression itself but those on the recognition of the facial expression. Accordingly, connection between the sentences seems unnatural in this paragraph. Furthermore, to tell that the participants did not get enough sleep, a questionnaire should have been conducted or convincing reasons for the inference should be presented.
Author Response
I have attached a file containing the response to Reviewer 1

Reviewer 2 Report
The paper presents a study that compares facial expression identification of healthy and patient groups by AI-based techniques versus human examiners.
While the paper is interesting, the study has several limitations that must be addressed.
1- Authors mentioned that they used MTCNN for facial expression detection. This isn't very clear because MTCNN is an AI model for face detection, not face expression detection, and there is a big difference. MTCNN can detect an image's face(s) but can't tell about emotions. Emotions are detected by other deep learning models, also known as sentiment analysis. This was not mentioned in this paper.
2- The results reported have limitations in the sense that the agreement between human observers and the AI model (mentioned as MTCNN) was analyzed for happy faces, but it should be analyzed for all sentiments.
Second, the accuracy of the AI model may depend on several factors (hyperparameter tuning), which were not mentioned or discussed in this paper. In summary, the AI model for sentiment analysis but much more elaborated, and experiments must be reported so that the reader can understand the settings of the study. The section describing MTCNN is superficial and does not present sufficient technical background to understand the models used and their performance. At this stage, the results are limited and can't be considered as to be generalized for the use case mentioned.
Author Response
I have uploaded the file containing the response to Reviewer 2

Round 2
Reviewer 1 Report
Based on the comments, the authors revised the manuscript well. Now this manuscript has a minor issue to be addressed.
As they mention in the method section, the authors used a system that integrates MTCNN-based face detection and DCNN-based expression recognition models. Since MTCNN is not a network for facial emotion recognition (FER), the phrase “MTCNN-based FER” is somewhat misleading and should be replaced with other appropriate words in the manuscript including the title.
Author Response
I have attached a file containing the response to reviewers
